# The Gut Microbiome’s Impact on the Pathogenesis and Treatment of Gastric Cancer—An Updated Literature Review

**DOI:** 10.3390/cancers17172795

**Published:** 2025-08-27

**Authors:** Ahmed S. Mohamed, Ruchi Bhuju, Emelyn Martinez, Marina Basta, Ashrakat Deyab, Charlene Mansour, Daniel Tejada, Vishal Deshpande, Sameh Elias, Vignesh Krishnan Nagesh

**Affiliations:** Department of Internal Medicine, Hackensack Palisades Medical Center, North Bergen, NJ 07047, USA; ruchibhuju3@gmail.com (R.B.); emelyn03.em@gmail.com (E.M.); msambasta95@gmail.com (M.B.); ashrakatdeyab@gmail.com (A.D.); cm1621@njms.rutgers.edu (C.M.); dt688@scarletmail.rutgers.edu (D.T.); sameh.elias@hmhn.org (S.E.)

**Keywords:** gastric cancer, gut microbiome, *Helicobacter pylori*, carcinogenesis, microbial biomarkers, microbiome-targeted therapy

## Abstract

Gastric cancer (GC) remains a major global health challenge, with the gut microbiome increasingly recognized as a critical factor in its pathogenesis and treatment outcomes. This review synthesizes recent evidence on how microbial communities—both gastric and intestinal—interact with known carcinogenic factors, notably *Helicobacter pylori*, and influence GC risk, progression, and therapy response. Special emphasis is placed on human post-eradication microbiome studies, revealing that *H. pylori* clearance reshapes gastric and intestinal microbial networks in ways that may have lasting implications for host immunity and carcinogenesis. We also discuss microbial biomarkers with potential clinical utility in GC risk stratification and highlight translational opportunities for microbiome-targeted interventions, including probiotics, dietary modulation, and microbiota-informed personalized therapy. Key gaps are identified, particularly the need for longitudinal, multi-omic profiling in diverse populations and the integration of microbiome data into GC screening and management. This synthesis provides clinicians and researchers with a current framework for understanding microbiome–GC interactions and guiding future research priorities.

## 1. Introduction

Gastric cancer is a significant global health concern, ranking as the fifth most common cancer worldwide and the third leading cause of cancer-related mortality [1]. The disease is categorized by anatomical subsites (cardia and non-cardia) and histological types (intestinal and diffuse), each exhibiting distinct epidemiological patterns [2]. 

In 2020, researchers conducted a global study on gastric cancer (GC), analyzing data from 34 countries. The study specifically examined records from the previous decade, providing a detailed review of trends and patterns between 2010 and 2019, and predicts that the incidence of GC is on a decreasing trend in the majority of countries, including high-incidence countries such as Japan (age-standardized rate (ASR): 36 in 2010 vs. ASR 30 in 2035) but also low-incidence countries such as Australia (ASR 5.1 in 2010 vs. ASR 4.6 in 2035). A total of 16 countries is predicted to fall below the rare disease threshold (defined as 6 per 100,000 person-years) by 2035 [3]. Geographically, the highest incidence rates are observed in high-income Asia Pacific countries (29.5 per 100,000) and East Asia (28.6 per 100,000), with China accounting for nearly half of global cases. The aging population in China contributes to the sustained burden of gastric cancer, as older adults (aged ≥ 60 years) exhibit higher mortality rates, with stomach cancer ranking as the second leading cause of cancer-related deaths in this age group [4]. In contrast, regions like sub-Saharan Africa and high-income North America report the lowest rates [5]. The global age-standardized incidence and mortality rates of gastric cancer declined from 22.44 and 20.48 per 100,000 in 1990 to 15.59 and 11.88 in 2019, respectively. According to the Global Burden of Disease (GBD) 2021 study, age-standardized disability-adjusted life year (ASDLY) rates for gastric cancer fell from 886.24 to 358.42 per 100,000 between 1990 and 2030 [6,7]. For young-onset gastric cancer (under 50 years), global incidence decreased from 2.20 per 100,000 (95% uncertainty interval (UI): 2.04–2.36) in 1990 to 1.65 (95% UI: 1.52–1.79) in 2019, with an AAPC of −0.95 (95% CI: −1.25 to −0.65, *p* < 0.001) [6]. The age-standardized mortality rate also declined from 20.5 (19.2–21.6) per 100,000 in 1990 to 11.9 (10.8–12.8) in 2019. Among ages 15–40, the age-standardized death rate (ASDR) decreased from 493.4 (463.7–523.7) per 100,000 in 1990 to 268.4 (245.5–290.6) in 2019 [1]. According to the latest report from the International Agency for Research on Cancer (IARC), there were approximately 19.96 million new cancer cases and 9.74 million cancer-related deaths globally in 2022, of which new GC cases were 966,000, with nearly 660,000 deaths [8]. Among these, the number of new gastric cancer cases was 966,000, with nearly 660,000 deaths [8]. 

The most significant risk factor for GC is a positive family history of gastric cancer, followed by *Helicobacter pylori* infection (*H. pylori*) [9]. *H. pylori* infection is identified as the leading risk factor in East Asian countries, where strains of *H. pylori* exhibit stronger carcinogenic effects compared to European strains [10]. For the *H. pylori*-infected population, the adjusted cumulative risk of GC from birth to age 85 was significantly higher than that of the non-infected population [10]. Dietary and behavioral factors, such as high intake of salty or high-risk foods (frequent intake of beef and canned, smoked, and salted food and less frequent intake of fresh fruit/vegetables), alcohol consumption, smoking, and obesity, also contribute [4,9]. Young-onset gastric carcinoma cases have a slight female predominance over males [9]. Clustering of multiple risk factors is more common in males, with 58.5% having at least two risk factors compared to 36.8% of females [11]. Notably, individuals with three or more risk factors were significantly less likely to adhere to screening guidelines [11]. Familial aggregation of gastric cancer is observed in approximately 10% of cases, though less than 3% are attributed to hereditary genetic mutations such as those in CDH1, CTNNA1, and APC [12]. One of the primary challenges in gastric cancer management is the lack of widespread screening programs, particularly in low-resource settings where the disease burden is highest [12]. There are disparities in screening adherence, particularly among high-risk individuals with multiple lifestyle risk factors. Barriers include lack of self-care, inadequate knowledge about cancer risk, and insufficient recommendations from healthcare providers [11,13]. While family history serves as a simple tool for risk stratification, its utility is limited by the reliance on self-reported data, which may introduce misclassification bias [14,15,16].

Dietary fiber has emerged as a potential protective factor against GC. A pooled analysis of 11 case–control studies within the Stomach Cancer Pooling (StoP) project demonstrated an inverse association between fiber intake and GC risk, with an odds ratio (OR) of 0.72 (95% CI: 0.59–0.88) for the highest versus the lowest quartile of fiber consumption [2]. 

To compile the evidence presented in this review, a comprehensive search of the scientific literature was conducted across major databases, including PubMed, Scopus, and Web of Science. Relevant studies were identified using a combination of keywords related to *Helicobacter pylori*, gastric microbiota, gastric cancer, probiotics, fecal microbiota transplantation, and microbiome-targeted therapies. Priority was given to recent studies, including randomized controlled trials, cohort studies, and high-quality reviews, while ensuring inclusion of seminal work that has shaped the understanding of microbial influences on gastric carcinogenesis. Articles that were not in English, lacked sufficient methodological detail, or focused solely on unrelated gastrointestinal conditions were excluded. Studies were critically appraised for relevance to human and animal models of gastric cancer, as well as mechanistic studies exploring microbiome–host interactions. This approach aimed to synthesize current knowledge comprehensively while maintaining a focus on clinically and mechanistically significant findings.

## 2. Role of Gut Microbiota and Gastric Carcinogenesis

### 2.1. Microbial Dysbiosis in Gastric Cancer

Studies have shown that the microbial composition of gastric cancer patients differs significantly from that of healthy individuals. In gastric cancer, microbial diversity is typically reduced (i.e., lower alpha diversity), and the composition shifts toward genera that are more pro-inflammatory or metabolically carcinogenic [17,18].

In particular, there is a higher abundance of *Streptococcus*, *Lactobacillus*, *Fusobacterium*, and *Peptostreptococcus* in the stomachs of GC patients, along with a decline in normal commensals such as *Prevotella* and *Neisseria* [19]. These alterations persist even in *H. pylori*-negative cases, suggesting a broader role of non-*H. pylori* microbiota in gastric tumorigenesis. One study utilizing 16S rRNA sequencing of gastric mucosal samples found that non-*H. pylori* bacteria not only colonized gastric tissues more densely in cancer patients but were also metabolically active in ways that supported carcinogenic processes [17].

### 2.2. Inflammatory Pathways and Immune Modulation

A fundamental mechanism by which gut microbiota contributes to carcinogenesis is through chronic inflammation. In a healthy host, the immune system and microbiota maintain a symbiotic equilibrium [20]. However, dysbiosis disrupts this balance, often leading to the activation of inflammatory signaling pathways such as NF-κB and STAT3 [21]. These transcription factors are well-known to regulate genes that control cell proliferation, apoptosis inhibition, angiogenesis, and DNA repair [21].

Microbial components such as lipopolysaccharides (LPS), flagellins, and peptidoglycans can interact with pattern recognition receptors like Toll-like receptors (TLRs) on gastric epithelial cells [22]. This interaction stimulates the production of pro-inflammatory cytokines, including IL-1β, IL-6, and TNF-α (Figure 1). Persistent exposure to these cytokines creates a pro-tumorigenic microenvironment by inducing DNA damage, promoting epithelial-to-mesenchymal transition, and encouraging immune evasion [21,23].

### 2.3. Carcinogenic Metabolites

Gut microbes produce a wide array of metabolites that can influence host cellular function. Some of these microbial products are implicated in carcinogenesis. For instance, nitrosating bacteria such as *Neisseria*, *Haemophilus*, and *Actinomyces* can convert dietary nitrates into carcinogenic N-nitroso compounds (NOCs), which are directly mutagenic and have been linked to gastric and esophageal cancers [19]. Other bacterial metabolites, such as reactive oxygen species (ROS) and short-chain fatty acids (SCFAs), can also play a role. While some SCFAs like butyrate are generally considered anti-inflammatory in the colon, in the acidic gastric environment, lactic acid produced by overabundant *Lactobacillus* species has been shown to lower local pH further and promote hypoxic conditions that favor tumor growth (Figure 1) [17,25]. Moreover, bacterial-derived toxins such as cytolethal distending toxin (CDT) and colibactin can induce direct DNA damage in host cells, leading to genomic instability, a hallmark of cancer [21,23].

Gut dysbiosis resulting from risk factors such as diet, obesity, and genetics promotes the production of pro-carcinogenic metabolites (e.g., NOCs, secondary bile acids, ROS) and disrupts epithelial integrity. These events activate chronic inflammation, immune evasion, and oncogenic signaling pathways such as Wnt/β-catenin and PI3K/Akt, ultimately facilitating tumorigenesis (Figure 1). However, it is important to note that most evidence stems from cross-sectional studies, animal models, or in vitro experiments [18,20,21], which limits the ability to establish direct causal relationships. Longitudinal cohort studies tracking microbiome changes over time, before and after *H. pylori* eradication, are still limited, and host genetic background, which can influence microbiome structure and individual susceptibility to gastric carcinogenesis, remains an underexplored factor [17,22]. Future research incorporating these elements, along with well-designed human interventional studies, is essential to clarify the role of specific microbial metabolites in gastric cancer pathogenesis.

### 2.4. Host–Microbiota Gene Interactions

Beyond inflammation and metabolites, recent research has explored how host gene expression is altered by microbial activity. Microbiota can influence signaling pathways such as Wnt/β-catenin and PI3K/Akt, which are frequently activated in gastric cancers [26] (Figure 1). In one study, changes in the microbial community were closely associated with methylation patterns of tumor suppressor genes in gastric tissue, suggesting epigenetic regulation as another pathway through which microbiota can promote carcinogenesis [23].

Moreover, bacterial biofilms found in the stomach have been shown to induce a hypoxic and inflammatory state that activates hypoxia-inducible factors (HIFs) and promotes angiogenesis—another cancer hallmark [19]. Importantly, host genetic background has been recognized as a key determinant shaping microbiome structure and influencing gastric cancer susceptibility. Variations in genes related to innate immunity, such as TLRs, NOD-like receptors, and cytokine regulators, can modulate microbial colonization patterns, immune responses, and the inflammatory tone of the gastric mucosa [24,25]. For instance, polymorphisms in TLR2 and TLR4 are associated with altered responses to *H. pylori* and differential microbial community profiles, which can affect progression from chronic gastritis to intestinal metaplasia and adenocarcinoma [25]. Similarly, host genetic differences affecting mucosal barrier function and epithelial signaling can create niches that favor specific bacterial genera, thereby influencing microbial metabolite production, dysbiosis severity, and ultimately cancer risk [24,26]. These findings underscore that gastric carcinogenesis is a result of a complex interplay between host genetics, microbial composition, and environmental factors, highlighting the potential of personalized approaches targeting both host and microbial factors for prevention and therapy [24,25,26].

### 2.5. Animal Model Evidence

Animal models have provided experimental support for these mechanisms. In a study using rats treated with N-methyl-N′-nitro-N-nitrosoguanidine (MNNG) to induce gastric cancer, significant changes in gut microbial composition were observed during the cancer progression phase. Specifically, a reduction in microbial diversity and a shift toward *Lactobacillus* and *Bacteroides* were noted, mirroring findings in human GC patients [18].

The gut microbiota is an active participant in gastric carcinogenesis, contributing far beyond the scope of *H. pylori*. Through inflammation, immune modulation, genotoxic metabolites, and alterations in host signaling and gene expression, microbial communities help shape the tumor microenvironment. Dysbiosis is not just a consequence of gastric cancer but a potential driver of its initiation and progression. Understanding the specific microbial signatures and mechanisms underlying this process will be essential for early detection, risk stratification, and future research into preventive strategies.

## 3. Impact of *Helicobacter pylori* and Beyond

*H. pylori* is a Gram-negative, spiral-shaped bacterium that colonizes the gastric epithelium and is the most prominent risk factor for gastric cancer [27]. It infects over half of the world’s population, though most carriers remain asymptomatic [28]. In a fraction of cases, however, infection progresses to serious gastric complications such as peptic ulcers, mucosa-associated lymphoid tissue (MALT) lymphoma, and gastric adenocarcinoma [29]. Due to its established role in gastric tumorigenesis, the International Agency for Research on Cancer classifies *H. pylori* as a Group 1 carcinogen [2]. More recently, investigations have expanded beyond the stomach, identifying *H. pylori* as a contributor to extragastric disorders, including neurologic, cardiovascular, metabolic, and dermatologic conditions [30].

### H. pylori and Gastric Carcinogenesis

The carcinogenic potential of *H. pylori* is mediated by persistent inflammation, direct epithelial injury, and complex interactions with host cellular pathways [31]. Chronic infection leads to gastritis, generating reactive oxygen and nitrogen species that inflict DNA damage on gastric epithelial cells [27]. This pro-inflammatory microenvironment promotes aberrant cellular proliferation, inhibits apoptosis, and creates genomic instability—all of which facilitate neoplastic transformation [32].

A key mechanism of *H. pylori* pathogenicity lies in its virulence factors, most notably cytotoxin-associated gene A (CagA) and vacuolating cytotoxin A (VacA) [33]. CagA is translocated into gastric epithelial cells via a type IV secretion system, where it becomes phosphorylated and hijacks host signaling pathways, including SHP-2, ERK, and Wnt/β-catenin. These changes promote cellular elongation, loss of polarity, and carcinogenic signaling [33]. VacA, in contrast, targets mitochondria, induces vacuolization, and suppresses immune responses by interfering with antigen presentation [34].

*H. pylori* also contributes to epigenetic modifications in the gastric mucosa, leading to silencing of tumor suppressor genes such as E-cadherin and p16 through aberrant DNA methylation [12]. Moreover, it alters the expression of host microRNAs that regulate apoptosis, proliferation, and immune evasion, thereby reinforcing the malignant phenotype [31].

Although *H. pylori* has long been considered the dominant organism in the gastric niche, recent metagenomic analyses reveal a diverse microbial community whose composition is significantly altered in the presence of *H. pylori*. Colonization by *H. pylori* leads to reduced microbial diversity (alpha diversity) and enrichment of pro-inflammatory and nitrosating bacteria. These bacteria convert nitrates into N-nitroso compounds, known carcinogens that can further promote mutagenesis [35]. Microbial dysbiosis also increases the abundance of lactic acid-producing bacteria, which lower pH and generate a hypoxic environment favorable to cancer development [29].

Mouse studies provide mechanistic support for this synergy: germ-free mice colonized solely with *H. pylori* develop less severe gastric pathology compared to those harboring both *H. pylori* and a conventional microbiota derived from specific pathogen-free mice, representing the full gastrointestinal microbial community rather than gastric bacteria alone. This suggests that the pathogenicity of *H. pylori* is amplified by interactions with the broader gut microbiota [27].

Beyond its role in gastric disease, *H. pylori* infection is associated with a variety of extragastric conditions, likely mediated by systemic inflammation, molecular mimicry, or alterations in the gut–brain or gut–immune axis [30]. In neurology, epidemiological studies suggest a correlation between *H. pylori* and neurodegenerative disorders such as Alzheimer’s disease and Parkinson’s disease. Proposed mechanisms include chronic inflammation leading to blood–brain barrier disruption, molecular mimicry, and direct neurotoxic effects of microbial metabolites. In cardiovascular medicine, *H. pylori* has been implicated in the pathogenesis of atherosclerosis, myocardial infarction, and stroke. These effects are thought to be mediated by systemic inflammatory cytokines, increased oxidative stress, and endothelial dysfunction [36,37]. Metabolically, *H. pylori* infection is linked to insulin resistance and type 2 diabetes mellitus. The bacterium promotes the production of interleukin-6 (IL-6) and tumor necrosis factor-alpha (TNF-α), both of which interfere with insulin signaling pathways in peripheral tissues [30].

*H. pylori* has also been associated with dermatologic diseases, including chronic urticaria, rosacea, and psoriasis. These associations are hypothesized to result from immune system activation, antigen mimicry, and release of vasoactive or inflammatory mediators [38]. *Helicobacter pylori* remains a major driver of gastric carcinogenesis through mechanisms involving inflammation, host cell signaling disruption, and epigenetic remodeling [31,32,33]. Its interactions with the gastric microbiome amplify its carcinogenic potential, and its role in extragastric diseases highlights the systemic consequences of chronic infection [29,30]. Given its widespread prevalence and diverse clinical manifestations, comprehensive strategies for early detection, eradication, and monitoring of *H. pylori* should be prioritized in cancer prevention and broader health frameworks [27,28].

## 4. Microbiota-Driven Inflammation and Cancer Progression

The gut microbiota is a dynamic and metabolically active community that influences host immunity, metabolism, and tissue homeostasis [39]. Disruption of this microbial equilibrium, known as dysbiosis, has been implicated in a range of chronic inflammatory disorders, including cancer [40]. A growing body of literature supports that persistent microbiota-driven inflammation can facilitate tumor initiation, progression, and immune evasion [41]. This inflammation-centered paradigm of tumorigenesis has drawn attention to the gut microbiota not merely as a bystander but as an active contributor to cancer development.

The intestinal epithelial barrier plays a vital role in segregating luminal microbes from the immune system [42]. Under conditions of dysbiosis, the barrier becomes more permeable due to downregulation of tight junction proteins, such as claudins and occludins [43]. This increased permeability allows microbial products like lipopolysaccharide (LPS) to translocate across the mucosa and activate inflammatory pathways [43]. These translocated products can stimulate macrophages and dendritic cells, promoting pro-inflammatory cytokine release and tissue damage [44].

Host recognition of microbial antigens occurs via pattern recognition receptors, notably Toll-like receptors (TLRs) and nucleotide-binding oligomerization domain (NOD)-like receptors [45]. Engagement of these receptors by microbial-associated molecular patterns leads to downstream activation of NF-κB and interferon signaling [46]. This results in the production of tumor-promoting cytokines like IL-6, TNF-α, and IL-1β, which support a microenvironment conducive to neoplastic transformation [47]. Moreover, TLR activation in epithelial cells can promote cell proliferation and inhibit apoptosis, further fueling tumorigenesis [46,47].

Microbiota composition heavily influences immune cell phenotypes and function [48]. Dysbiosis can skew T-cell populations toward pro-inflammatory Th17 responses while reducing regulatory T cells (Tregs) that ordinarily restrain inflammation [49]. This imbalance amplifies mucosal immune responses, sustaining low-grade inflammation over time [48,49]. Prolonged inflammatory signaling can enhance oxidative stress and DNA damage, both of which are early steps in carcinogenesis [49].

Certain gut bacteria metabolize dietary components into bioactive compounds that may be genotoxic or promote proliferation. For example, *Clostridium* species produce secondary bile acids like deoxycholic acid, which can induce DNA damage and promote colorectal tumor growth [50]. Other microbial metabolites, such as hydrogen sulfide, have been shown to impair DNA repair and facilitate mutagenesis [51]. Short-chain fatty acids (SCFAs) like butyrate, though typically anti-inflammatory, may paradoxically fuel tumor cell proliferation under hypoxic or oncogenic conditions [52] (Table 1).

Understanding how microbiota-driven inflammation contributes to cancer opens up new therapeutic possibilities. Probiotic and prebiotic interventions may help restore microbial balance and reduce inflammation [58]. Fecal microbiota transplantation (FMT) is being explored not only for gastrointestinal diseases but also as an adjunct to cancer immunotherapy [59]. Moreover, targeting microbial metabolites or their host signaling pathways may yield novel chemopreventive or therapeutic agents. The microbiota’s influence on cancer extends far beyond passive association—it actively shapes the inflammatory and metabolic landscape in which tumors arise and evolve. Through barrier disruption, immune modulation, and the production of carcinogenic metabolites, dysbiotic microbial communities can sustain chronic inflammation that drives carcinogenesis. Integrative strategies targeting these pathways may pave the way toward innovative prevention and treatment approaches [60].

## 5. Post-*Helicobacter pylori* Eradication and Gastric Cancer Risk

When present, *Helicobacter pylori* dominates the gastric microbiota by greatly increasing its relative abundance. Its capacity to survive and proliferate in the acidic stomach environment gives it a competitive advantage, resulting in reduced diversity among other bacterial populations. Studies have consequently shown that *Helicobacter pylori* infection leads to a marked decline in overall microbial diversity, with commensal bacterial groups such as Firmicutes, Bacteroidetes, and Actinobacteria becoming less abundant [53,61] (Table 1).

A diverse and balanced gut microbiome is crucial for both gastric and systemic health, influencing a wide array of physiological processes. It plays a fundamental role in digestion and nutrient absorption by breaking down complex carbohydrates and extracting nutrients that the human body cannot digest independently. Additionally, the gut microbiota contributes to the synthesis of vital compounds, such as vitamins, and supports the integrity of the gastrointestinal barrier. By maintaining this barrier, the microbiome prevents the movement of harmful substances into the bloodstream, which then reduces systemic inflammation [62].

Beyond digestive functions, the gut microbiota is important in modulating the immune system. It aids in the development and maturation of immune responses, enabling the body to differentiate between pathogenic and beneficial microbes. Beneficial bacteria also competitively inhibit pathogen colonization by occupying niches and resources, thereby preventing infections. Furthermore, the microbiome influences host metabolism, including energy balance, fat storage, and glucose regulation [62].

Dysbiosis, as discussed previously, is increasingly recognized as a contributor to chronic inflammation.

Eradication of *Helicobacter pylori* leads to notable gastric and intestinal dysbiosis, with implications for host health. Post-eradication, the gastric microbial environment undergoes significant changes. In a study that summarized multiple longitudinal cohorts, it was noted that *Enterococcus* spp. and other opportunistic taxa increase within 1–3 months after eradication, with partial reversion toward baseline by 6–12 months. Several gastric mucosal genera, including *Actinomyces*, *Granulicatella*, *Parvimonas*, *Peptostreptococcus*, *Prevotella*, *Rothia*, *Streptococcus*, *Rhodococcus*, and *Lactobacillus*, have been implicated in the development of precancerous lesions and may proliferate following eradication, potentially influencing gastric cancer risk. Studies have reported a decrease in Actinobacteria within the gastric mucosa, while Proteobacteria levels often rise transiently during short-term follow-up before returning to baseline [57] (Table 1). Enterobacteriaceae and Enterococcus species also tend to increase shortly after eradication, potentially contributing to temporary dysbiosis [54] (Table 1). Similarly, a long-term prospective cohort study of 93 subjects assessed gastric corpus microbiota at baseline, 6 months, 1 year, and up to 5 years post-eradication. They found a marked short-term rise in *Proteobacteria* and *Enterobacteriaceae*, with *Actinobacteria* significantly reduced even at the 5-year follow-up [55] (Table 1).

At the intestinal level, eradication therapy has been associated with an increase in *Proteobacteria* and a reduction in beneficial phyla such as *Bacteroidetes* and *Actinobacteria* [57]. In several follow-up cohorts, these changes were evident within weeks of eradication and, in the case of *Actinobacteria*, persisted for more than 6 months. Such persistent shifts may contribute to long-term dysbiosis, adverse gastrointestinal symptoms, and chronic inflammation. Together, these cohort findings suggest that eradication not only removes a key gastric pathogen but also remodels both gastric and gut microbial communities in ways that may have lasting implications for carcinogenesis and host immunity [57] (Table 1). Additionally, eradication can affect microbial metabolic functions in both the stomach and intestine, further disrupting host–microbe interactions. These findings suggest that while *Helicobacter pylori* eradication is essential in certain clinical contexts, such as peptic ulcer disease or gastric malignancy risk, it should not be universally applied. The decision to pursue eradication should consider the potential for microbiota disruption and weigh the benefit–risk ratio on an individual basis [55] (Table 1).

The connection between altered microbiota and cancer pathways, particularly in the context of *Helicobacter pylori* (*H. pylori*) infection and eradication, centers around inflammation, epithelial damage, and dysbiosis-driven carcinogenesis. *Helicobacter pylori* is recognized as a Class I human carcinogen due to its role in triggering Correa’s cascade, a multistep process that transforms normal gastric mucosa into adenocarcinoma via chronic gastritis, atrophic gastritis, intestinal metaplasia, and dysplasia. The infection induces chronic inflammation, generating reactive oxygen and nitrogen species that promote DNA damage and genomic instability [56].

However, *Helicobacter pylori* eradication does not always reverse these carcinogenic processes, especially in patients who already exhibit pre-neoplastic lesions. Following eradication, the gastric and intestinal microbiota often undergo significant compositional changes, causing dysbiosis characterized by a decrease in protective microbial populations (such as Actinobacteria) and an increase in potentially pathogenic ones (e.g., Proteobacteria, Enterobacteriaceae) that perpetuates mucosal inflammation and maintains a microenvironment that is carcinogenic. Some of these bacterial genera (such as *Streptococcus*, *Prevotella*, and *Lactobacillus*) have been associated with gastric mucosal changes that resemble precancerous conditions [54]. Moreover, the loss of microbial diversity and beneficial metabolites post-eradication may impair the mucosal barrier and immune regulation, fostering conditions that allow oncogenic mutations and epigenetic alterations to persist or advance [56].

Thus, while *Helicobacter pylori* eradication has been shown to reduce inflammation and halt early molecular damage in many patients, it may also give rise to a new microbiota composition with carcinogenic potential, especially if pre-existing histologic changes are present [54]. This highlights the importance of both microbial and histologic assessment when determining long-term cancer risk and supports the integration of microbiota-targeted strategies, such as probiotics or microbiome-modulating therapies, in the surveillance and management of high-risk individuals post-eradication.

Future research should aim to find the specific microbial and metabolic pathways involved in cancer risk, enabling more personalized and microbiota-driven strategies for the prevention and management of gastric cancer.

As we know, *Helicobacter pylori* infection and a family history of gastric cancer are the main risk factors for gastric cancer. A study involving individuals with a family history of gastric cancer found that *Helicobacter pylori* eradication significantly reduced the risk of developing gastric cancer, though it did not eliminate the risk entirely. Participants who received treatment had a 55% lower incidence of gastric cancer compared to those given a placebo, and the reduction reached 73% among those in whom eradication was successfully achieved. These findings are in line with results from earlier meta-analyses and long-term trials, which consistently show a substantial risk reduction following eradication therapy [63]. However, the occurrence of gastric cancer even after successful treatment suggests that other factors, such as genetic susceptibility or residual gastric damage, may continue to contribute to cancer risk independent of *Helicobacter pylori* status [63].

The study also highlighted that treatment failure limits the benefit of eradication, as individuals with persistent *Helicobacter pylori* infection had cancer risks comparable to those who received no treatment. This reinforces the importance of confirming eradication success, as emphasized by the “test–treat–test” strategy [63]. Interestingly, the incidence of gastric adenoma did not differ between treated and untreated groups, suggesting that the carcinogenic pathway related to *Helicobacter pylori* may not involve the classic adenoma–carcinoma sequence. These findings support the use of eradication therapy as a preventive measure, especially in high-risk populations, but also highlight the ongoing need for surveillance and early detection to further reduce gastric cancer burden [63].

Gastric intestinal metaplasia is a potential reversible outcome of injury and repair, rather than a lesion directly linked to carcinogenesis. It serves more as a biomarker of past gastric damage than as a definitive precursor to cancer. The risk of gastric cancer is better assessed by evaluating the severity, extent, and especially the cause of atrophic changes in the mucosa [64]. Even after *Helicobacter pylori* eradication, residual cancer risk can persist due to irreversible alterations like atrophy and intestinal metaplasia [65]. These mucosal changes are part of a well-known sequence of disease progression, commonly referred to as the “Correa cascade”, which outlines the gradual transformation from normal mucosa to malignancy [64].

Although the original model placed intestinal metaplasia just before cancer in the sequence, it is now understood that this does not imply a direct path to malignancy. Still, the presence of such changes reflects a background of chronic inflammation and structural damage that may not completely reverse after eradication [64]. The misconception that intestinal metaplasia is inherently pre-neoplastic has caused clinical uncertainty and patient anxiety when it appears in biopsy reports [64]. In practice, these findings indicate a field of altered tissue that, while not immediately dangerous, may still justify ongoing monitoring in some patients even after the underlying infection is treated.

Residual risk of gastric cancer persists due to a combination of genetic, inflammatory, and environmental factors [66]. Genetic mutations, such as those involved in hereditary gastric cancer syndromes like Hereditary Diffuse Gastric Cancer (HDGC), caused by CDH1 gene mutations, for example, significantly elevate cancer risk. Even in the absence of a specific genetic syndrome, a strong family history of gastric cancer can signal a higher risk [67]. Chronic inflammation is another major contributor, and *Helicobacter pylori* infection is a well-established cause, linked to a large percentage of cases. Other sources of inflammation, such as Gastro-Esophageal Reflux Disease (GERD), autoimmune gastritis, and various chronic gastric conditions, also play a role in carcinogenesis [66].

Environmental cofactors further compound the risk. Diets rich in salt, smoked, or preserved foods, along with smoking and heavy alcohol consumption, are all known to increase susceptibility to gastric cancer [68]. Occupational exposure to harmful substances like industrial chemicals or dusty environments, and the presence of obesity, which are all often associated with chronic low-grade inflammation, add yet another layer to risk [68]. Together, these elements underscore the multifaceted nature of residual gastric cancer risk, even in populations with declining incidence rates due to improved diagnostics and preventive measures [66].

Meta-analyses have consistently shown that *Helicobacter pylori* eradication is associated with a reduced risk of gastric cancer. Individuals who undergo eradication therapy have a significantly lower risk compared to those who remain infected. One meta-analysis reported a relative risk of 0.65, indicating a 35% reduction in cancer risk, while another found a risk ratio of 0.54 for incidence and 0.61 for mortality [65]. These findings support the role of *Helicobacter pylori* treatment as an effective preventive strategy against gastric cancer.

Epidemiological evidence shows that the risk of gastric adenocarcinoma persists for years following *Helicobacter pylori* eradication, demonstrating a distinct time-sensitive pattern. A comprehensive population-based cohort study conducted across five countries, Denmark, Finland, Iceland, Norway, and Sweden, provides crucial insight into this temporal risk trajectory. The study included over 659,000 adults treated for *Helicobacter pylori* infection between 1995 and 2019, with follow-up data spanning nearly 5.5 million person-years. Notably, the standardized incidence ratio (SIR) of gastric noncardia adenocarcinoma remained significantly elevated in the first 1–5 years post-eradication (SIR 2.27; 95% CI 2.10–2.44), underscoring a period of heightened vulnerability. Over time, however, the risk steadily declined, with the SIR approaching parity with the general population by 11–24 years after treatment (SIR 1.11; 95% CI 0.98–1.27) [69]. These findings suggest that although eradication reduces cancer risk in the long term, a residual risk persists for over a decade, highlighting the need for vigilant post-eradication surveillance during this critical window [69].

Despite the benefits of *Helicobacter pylori* eradication, a residual risk of gastric cancer persists, particularly in patients with established pre-neoplastic changes such as atrophic gastritis and intestinal metaplasia [65,69]. While early eradication prior to the onset of these lesions offers the greatest protective effect, eradication at later stages may only halt progression rather than reverse existing histological damage [69]. To address this ongoing risk, the Kyoto Global Consensus emphasizes the importance of endoscopic and histological evaluation in patients with non-invasively diagnosed *Helicobacter pylori* infection and clinical indicators of mucosal atrophy, such as advanced age, a history of gastric ulcers, or abnormal serum pepsinogen levels [70]. In these high-risk individuals, personalized risk stratification models and validated staging systems can guide surveillance intervals and management strategies. Long-term endoscopic monitoring remains essential to enable early cancer detection and improve outcomes in this vulnerable population [70].

Although *Helicobacter pylori* eradication is a key strategy in reducing gastric cancer risk, surveillance remains important in patients with gastric intestinal metaplasia (GIM), particularly those with histologic evidence of incomplete or extensive metaplasia [71]. These individuals face a significantly higher risk of progression to gastric cancer compared to those with complete or limited forms of intestinal metaplasia. Once GIM has developed, the effectiveness of *Helicobacter pylori* eradication in reversing mucosal alterations or preventing carcinogenic progression becomes uncertain, as such changes are often considered irreversible [72]. Furthermore, patients with atrophic gastritis or gastric atrophy continue to carry a substantial risk of gastric cancer even after *Helicobacter pylori* eradication, reinforcing the importance of ongoing surveillance [73]. Continued monitoring through endoscopic or non-endoscopic methods is recommended to evaluate for neoplastic progression and to enable timely clinical intervention. This is particularly crucial for individuals with a history of early, resectable gastric cancer, as the risk of developing metachronous lesions remains elevated despite successful eradication therapy. Although relatively uncommon, progression from GIM to gastric cancer is well documented, highlighting the need for vigilant follow-up [72]. Early detection through surveillance is essential because it enables diagnosis at a stage when gastric cancer is most treatable, thereby significantly improving patient outcomes and survival rates.

Current international guidelines for the surveillance of gastric intestinal metaplasia vary by region, reflecting differences in gastric cancer incidence and healthcare infrastructure. In Eastern countries such as Japan and South Korea, where gastric cancer prevalence is high, national screening programs are well established. In Japan, endoscopic or radiographic screening is recommended for all individuals aged 50 and older [72]. For patients with GIM, especially those with high-risk features such as incomplete or extensive metaplasia, a positive family history of gastric cancer, or lifestyle-related risks like smoking and alcohol use, surveillance endoscopy is typically advised every 1 to 3 years [72]. In contrast, Western guidelines adopt a more selective approach. The British Society of Gastroenterology and a 2019 European consensus statement recommend triennial endoscopic surveillance for patients with extensive GIM, particularly when it involves both the antrum and corpus. They also support surveillance for patients with localized GIM in the presence of risk factors such as incomplete histologic subtype, persistent *Helicobacter pylori* infection, family history of gastric cancer, or autoimmune gastritis [72]. In the United States, the American Gastroenterological Association recommends against routine surveillance in the general population. However, for patients with GIM who present with high-risk features (such as incomplete or extensive metaplasia, family history of gastric cancer, or being from high-incidence geographic or ethnic backgrounds), endoscopic surveillance may be considered at intervals of 3 to 5 years if clinically appropriate and patient-preferred [74].

Endoscopic surveillance practices are increasingly guided by advanced imaging techniques and quality benchmarks. The European Society of Gastrointestinal Endoscopy emphasizes the importance of high-quality upper gastrointestinal endoscopy, including proper time documentation, photo documentation, standardized reporting, and the use of high-resolution scopes. These practices aim to reduce the likelihood of missed lesions [74]. Virtual chromoendoscopy, utilizing technologies like narrow band imaging and blue light imaging, has become a key tool in visually identifying and grading GIM [74]. When GIM is detected endoscopically, targeted biopsies are essential to confirm the diagnosis, determine its extent (in both the antrum and corpus), and assess for *Helicobacter pylori* infection. If GIM is identified in both the antrum and corpus or if incomplete metaplasia is present, a 3-year surveillance interval is recommended. If there is any concern about the adequacy of the initial examination, a repeat endoscopy within one year is advised [74]. These guidelines reflect a growing consensus that surveillance should be tailored based on risk stratification, endoscopic findings, and histopathologic confirmation to ensure early detection, which is crucial for identifying gastric cancer at a stage when it is most treatable [74].

One of the current methods used to assess a patient’s risk of developing gastric cancer is the use of the OLGA (Operative Link on Gastritis Assessment) and OLGIM (Operative Link on Gastric Intestinal Metaplasia Assessment) staging systems. OLGA is a histological staging tool that evaluates gastric atrophy, incorporating both metaplastic and non-metaplastic components. In contrast, OLGIM focuses specifically on the extent and distribution of intestinal metaplasia. Both systems use a standardized five-stage scale (0 to IV) to classify histologic changes, with stages III and IV consistently associated with a significantly elevated risk of gastric cancer, thereby identifying individuals who may benefit from enhanced surveillance and preventive strategies [75].

Pepsinogen (PG) testing provides a noninvasive and cost-effective approach to identifying individuals at increased risk of gastric cancer, particularly after *Helicobacter pylori* eradication. By measuring fasting serum levels of PG-I and the PG-I/II ratio, both biomarkers of atrophic gastritis, a key step in the Correa cascade, PG testing can aid in the detection of precancerous gastric lesions [76]. This enables targeted endoscopic surveillance for high-risk individuals, a strategy especially valuable in low-incidence settings like the United States, where routine esophagogastroduodenoscopy (EGD) is not feasible for population-wide screening. Incorporating PG testing into primary care settings may improve early detection and reduce gastric cancer mortality, particularly in underserved and high-risk populations [76].

Gastrin, a hormone that regulates gastric acid secretion and mucosal proliferation, has also been implicated in gastric carcinogenesis. Elevated serum gastrin levels are often associated with conditions such as atrophic gastritis and intestinal metaplasia and may serve as a marker for increased cancer risk [77]. In particular, hypergastrinemia has been proposed as a predictor for the development of metachronous gastric cancer or secondary tumors occurring after initial curative treatment. Although long-term use of proton pump inhibitors (PPIs) can elevate serum gastrin levels, current evidence suggests that PPI-induced hypergastrinemia does not significantly increase the risk of metachronous gastric cancer [78].

New surveillance strategies, including AI-assisted endoscopy and microbiome profiling, are changing and improving the detection and management of gastric cancer (GC). AI-assisted endoscopy has significantly improved early detection and diagnostic accuracy, with deep learning models such as convolutional neural networks achieving sensitivities up to 92.2% in identifying GC lesions, surpassing traditional methods and even experienced endoscopists [79]. Furthermore, AI has shown high accuracy in detecting *Helicobacter pylori* infection and related precancerous gastric conditions, potentially minimizing unnecessary biopsies and enhancing personalized patient care [79]. Concurrently, microbiome profiling through 16S rRNA gene sequencing has identified a distinct dysbiotic gastric microbial community in GC patients, characterized by reduced microbial diversity and increased abundance of intestinal commensals [80]. This dysbiosis, quantified by a microbial dysbiosis index, effectively distinguishes GC from chronic gastritis and points to a genotoxic microbial environment that may play a role in gastric carcinogenesis [36].

Despite the significant reduction in gastric cancer (GC) risk following *Helicobacter pylori* eradication, several challenges remain in effective surveillance. Residual risk persists, particularly in patients with advanced mucosal changes such as atrophic gastritis or intestinal metaplasia, and a “field effect” may increase the likelihood of metachronous GC even after eradication [81]. Additionally, there is no consensus on optimal surveillance intervals, complicating long-term management. Endoscopic detection is also hindered post-eradication due to mucosal changes like patchy erythema and indistinct lesion borders, increasing the risk of misdiagnosis or delayed detection [80]. Furthermore, endoscopists’ lowered suspicion of GC after eradication may introduce detection bias. Limitations in current data include low event rates in prevention studies, regional variability in outcomes, and a need for further research to refine surveillance protocols, especially for patients with mild atrophic changes [65]. Finally, gastric cancers post-eradication exhibit diverse pathological features, encompassing both differentiated and undifferentiated types, which necessitate tailored surveillance approaches [82]. So, though *Helicobacter pylori* eradication remains a critical part of gastric cancer prevention, surveillance strategies must be optimized to address these persistent challenges and improve early cancer detection.

Stratified surveillance is essential for high-risk individuals following *Helicobacter pylori* eradication to improve early detection and survival outcomes in gastric cancer (GC). High-risk groups include those with a family history of GC, individuals from regions with high GC incidence (such as East Asia and South America), and patients with precancerous gastric conditions like atrophic gastritis or intestinal metaplasia [80]. Regular endoscopic surveillance, tailored to the patient’s risk profile and severity of mucosal changes, enables early identification of malignancies when less invasive treatments are most effective. Recommended surveillance intervals typically range from one to three years, with advanced imaging techniques applied as needed [80]. An individualized approach ensures optimal monitoring, balancing early detection with resource utilization and patient-specific risks.

Advances in technology, such as AI-assisted endoscopy and microbiome profiling, alongside emerging biomarkers, are promising areas in refining surveillance strategies for gastric cancer following *Helicobacter pylori* eradication [36,79]. These tools can improve early detection, enhance risk stratification, and personalize patient monitoring, particularly in high-risk groups where residual cancer risk persists [72,74]. However, the optimal timing and combination of surveillance modalities remain uncertain, which shows the need for continued research to establish evidence-based guidelines. Further studies will be crucial to validate these new innovations, optimize surveillance intervals, and ultimately improve outcomes for patients at risk of gastric cancer post-eradication [80].

## 6. Impact of Chemotherapy on the Gut Microbiome

Virtually every cytotoxic backbone used in gastric cancer perturbs the intestinal ecosystem, but the depth and durability of that injury vary with drug class, cumulative dose, and host factors. Large-scale next-generation sequencing shows a reproducible fall in α-diversity together with overgrowth of opportunists such as *Enterococcus* and *Streptococcus* after platinum/fluoropyrimidine cycles [83,84]. In a prospective solid-tumor cohort, this ecological shock reached a nadir at the onset of myelosuppression and normalized within six weeks in most patients [85]. Gastric cancer-specific data refine the picture: in a longitudinal FLOT registry, neoadjuvant chemotherapy produced only transient shifts, whereas the subsequent gastrectomy “locked in” an oralization pattern dominated by *Escherichia-Shigella* and *Veillonella*, suggesting that surgical anatomy, rather than chemotherapy per se, cements long-term dysbiosis [86].

The gut community is more than collateral damage—it actively conditions treatment response. In murine and ex vivo gastric cancer models, *Akkermansia muciniphila*–derived pentadecanoic acid restores oxaliplatin sensitivity by dampening tumor glycolysis [87]. Conversely, bacterial β-glucuronidases reconvert inactive SN-38G to toxic SN-38 in the colon, driving late-onset irinotecan diarrhea; structure-guided inhibitors abrogate this toxicity without blunting antitumor activity [88]. Clinically, low baseline diversity predicts severe gastrointestinal and hematological toxicity, whereas enrichment with *Bifidobacterium* correlates with higher pathological complete-response rates in peri-operative settings [87].

Intervention studies are moving from proof-of-concept to pragmatic trials. A 2022 meta-analysis of seventeen randomized studies reported a 35% relative-risk reduction in chemotherapy-induced diarrhea and oral mucositis with multi-strain probiotic blends, without compromising dose intensity [89]. Fecal microbiota transplantation (FMT) has now entered controlled pre-clinical pipelines: donor material from chemosensitive hosts restored oxaliplatin efficacy and attenuated cachexia in syngeneic gastric cancer mice, paving the way for first-in-human trials [90]. These benefits come with caveats—broad-spectrum antibiotics administered within four weeks of therapy initiation collapse commensal diversity and derail PD-1 blockade in advanced gastric cancer, a cautionary signal that likely extends to cytotoxics [91].

High-resolution metabolomics, gnotobiotic tumor models, and early-phase FMT/probiotic trials are converging on a future in which gut-directed adjuvants augment chemotherapy while blunting off-target toxicity. Two recent umbrella reviews emphasize that successful interventions must account for regional diet, surgery-induced anatomical change, and tumor-specific microbial niches; stratification by baseline microbial and metabolite signatures is therefore essential for forthcoming randomized studies in gastric cancer [19,92].

## 7. Emerging Therapeutic Perspectives

Recent randomized controlled trials (RCTs) and systematic reviews have started to address microbiome-targeted modulating therapies in the context of *Helicobacter pylori* and gastric cancer prevention and although this approach, especially the use of probiotics is not a new technology, the field still remains in early stages, with most data derived from pre-clinical studies, small clinical trials, or meta-analyses.

Probiotics have been the most extensively studied adjunct to *H. pylori* eradication regimens. The American College of Gastroenterology, in its 2024 guideline, notes that meta-analyses of RCTs (primarily from Asia) show a modest increase in eradication rates and a reduction in side effects when probiotics, especially Lactobacillus or multi-strain formulations, are added to standard therapy. These studies enrolled patients with chronic gastritis, peptic ulcer disease, or dyspepsia and sometimes included those with a history of failed eradication attempts. However, most probiotic trials are short-term, small-scale, and heterogeneous in strain composition and dosage, making it difficult to compare results or define optimal regimens. Many also lack rigorous blinding and do not adequately control for diet or concurrent antibiotic regimens, potentially confounding the observed benefits. Therefore, due to significant heterogeneity and lack of high-quality RCTs in North America, the society does not recommend routine use of probiotics as standard of care at this time but considers the evidence hypothesis-generating [93] (Table 2). A recent RCT also enrolled patients with biopsy-proven *H. pylori* and premalignant intestinal metaplasia who had successful eradication and randomized them to either receive probiotics or no treatment. Both groups underwent follow-up endoscopy to assess the regression of metaplasia. The underlying mechanism involves the ability of two probiotics, namely Lactobacillus acidophilus and Bifidobacterium lactis, to have anti-carcinogenesis effects to ameliorate the *H. pylori* Wnt/B-Canetin related COX-2 regression pathway by regulating the expression of microRNA. B. lactis decreased *H. pylori*-induced and nuclear β-catenin/COX-2 signaling and miR-185 expression in a dose-dependent manner. In the 6-month probiotic-treated patients had a significantly higher IM regression rate compared to the control arm (intention-to-treat: 37.5 vs. 11.5%, OR: 4.60, 95% CI: 1.134–18.65, *p* = 0.025). Patients without IM regression had significantly higher miR-185 levels in follow-up biopsies (*p*  <  0.01). There remains no role for microbiome-targeted therapies, including probiotics, in patients with established gastric cancer. However, future trials are expected in that population [94] (Table 2).

Prebiotics are defined as non-digestible food ingredients that selectively stimulate the growth or activity of beneficial bacteria in the gastrointestinal tract, thereby conferring health benefits to the host. In the context of *Helicobacter pylori* infection and gastric cancer, prebiotics are used to promote the proliferation of commensal bacteria (such as Bifidobacterium and Lactobacillus species), which may help restore microbial balance disrupted by *H. pylori* colonization or antibiotic therapy [95] (Table 2). Synbiotics are defined as a combination of probiotics and prebiotics, formulated to have a synergistic effect by enhancing the survival and activity of the probiotic organisms [96]. There have been a few meta-analyses of RCTs using synbiotics as adjuncts to eradication therapies. It was concluded in one study that there might be an increased eradication in combination with conventional therapies, together with decreased antibiotic adverse events. However, findings were statistically insignificant due to scarce available RCTs [97] (Table 2).

Fecal microbiota transplantation (FMT) involves the transfer of processed stool from a healthy donor into the gastrointestinal tract of a recipient, aiming to restore a balanced gut microbiome. Its established clinical use is for recurrent Clostridioides difficile infection, but its role in *Helicobacter pylori* infection and gastric cancer is investigational.

A recent RCT that was performed in China evaluated the use of FMT. A total of 30 patients were recruited and were given bismuth quadruple eradication therapy for 14 days, after which they were assigned to receive one-time FMT or placebo. The study found that while FMT did not accelerate restoration of gut microbiota diversity, it did alleviate gastrointestinal symptoms such as diarrhea in the short term. The gut microbiota generally returned to baseline by 10 weeks post-eradication, regardless of FMT, and no direct effect on *H. pylori* eradication rates was observed [98] (Table 2). For gastric cancer, however, a phase II RCT assigned patients with metastatic HER-2 negative gastric adenocarcinoma to allogenic FMT (healthy obese donors) vs. autologous FMT. Primary outcomes were to assess the effect of allogenic FMT on satiety. Secondary outcomes included disease control rate (DCR), overall survival (OS), progression-free survival (PFS), and toxicity. Although primary outcomes were not met, DCR was improved in the allogenic arm compared with the autologous group (*p* = 0.035), as well as OS of 365 versus 227 days [HR = 0.38; 95% confidence interval (CI), 0.14–1.05; *p* = 0.057] and PFS of 204 versus 93 days (HR = 0.50; 95% CI, 0.21–1.20; *p* = 0.092). Patients in the allogenic group showed a significant shift in fecal microbiota composition after FMT (*p* = 0.010), indicating proper engraftment of the donor microbiota. Despite promising findings, there still remains no established role of FMT in the treatment of gastric cancer, and larger FMT studies are needed [99] (Table 2). In summary, application of FMT to GC prevention or treatment remains highly investigational, with limitations including small sample sizes, short follow-up durations, donor variability, and uncertain long-term safety profiles. Importantly, many studies focus on intestinal rather than gastric microbial changes, leaving a gap in understanding direct gastric mucosal effects [98,99].

Personalized medicine approaches are methods that incorporate individual microbiome profiles as a form of microbiome-targeted therapy, and it involves using high-throughput sequencing and bioinformatics to characterize a patient’s unique gastric and gut microbial composition through biopsy samplings; identify specific dysbiotic patterns and microbial signatures associated with *H. pylori* persistence, gastric carcinogenesis, or treatment resistance; and then select or design interventions (potentially including targeted microbiome target therapies and dietary modifications to address those individual features [19,100]. One of the most recently developed tests is the Resident Gastric Microbiota Dysbiosis Test (RGM-DT). This test develops a score based on the relative abundance of specific bacterial taxa, such as increased bacillus in the antrum and decreased Rhizobiales, Weeksellaceae, and Veillonella (in the corpus). These changes are associated with the progression of atrophic gastritis to high-grade dysplasia and gastric cancer. RGM-DT demonstrated a very high specificity (88.9%), supporting them as predictive biomarkers for the neoplastic progression of atrophic gastritis [101]. Additional studies have identified other taxa relevant to gastric carcinogenesis, including Fusobacterium, Parvimonas, Peptostreptococcus, Prevotella, Streptococcus anginosus, Slackia exigua, and Dialister pneumosintes, which are abundant in gastric cancer, while other commensals such as Bifidobacterium, Bacillus, and Blautia are depleted [101]. This approach differs from traditional risk stratification, which is mainly based on demographic, clinical, and histopathologic factors such as age, family history, country of origin, *H. pylori* status, and presence of premalignant lesions. The American Gastroenterological Association recommends individualized risk assessment using these factors; however, it still does not incorporate microbiome profiling into standard practice. Traditional methods rely on endoscopic and histologic evaluation, serologic testing, and clinical risk factors, as opposed to direct assessment of the gastric microbial community [102,103].

**Table 2 cancers-17-02795-t002:** Microbiome-targeted interventions for modulating gastric cancer risk and study types, findings, and evidence level.

Intervention	Type of Study	Patient Population	Key Findings	Trial Phase	Evidence Level	Notes
Probiotics (Lactobacillus, multi-strain) [93]	Meta-analyses of RCTs	Chronic gastritis, peptic ulcer, dyspepsia, and some with failed eradication	Modest increase in *H. pylori* eradication rates, reduced side effects; heterogeneous data, mostly Asian studies	Meta-analyses (RCT)	Evidence-based (hypothesis-generating)	Not routine standard care in North America due to heterogeneity and lack of high-quality RCTs
Probiotics (L. acidophilus, B. lactis) [94]	RCT	*H. pylori* with premalignant intestinal metaplasia post-eradication	Higher regression of intestinal metaplasia (IM) (37.5% vs. 11.5%, OR 4.60, *p* = 0.025); miR-185 levels correlated with response	Phase II RCT	Investigational	Mechanistic support via COX-2/β-catenin pathway regulation; no current role in established gastric cancer
Prebiotics [95]	Review/Conceptual	General (in *H. pylori* infection context)	Promote growth of beneficial bacteria; no direct clinical trial data mentioned	Pre-clinical	Investigational	Potential to restore microbial balance; clinical trials needed
Synbiotics (probiotic + prebiotic) [97]	Meta-analyses of RCTs	*H. pylori* eradication adjunct	Possible increased eradication and decreased adverse events; statistically insignificant due to limited data	Meta-analyses (RCT)	Investigational	More RCTs needed to confirm effects
Fecal microbiota transplantation (FMT) [98]	RCT	*H. pylori* patients after bismuth quadruple therapy	No effect on eradication rates; alleviated short-term GI symptoms; microbiota returned to baseline by 10 weeks	Phase II RCT	Investigational	Small sample (30 pts); no acceleration of microbiota restoration
FMT (allogenic vs. autologous) [99]	RCT	Metastatic HER-2-negative gastric adenocarcinoma	Improved disease control rate (DCR), overall survival (OS), and progression-free survival (PFS) with allogenic FMT	Phase II RCT	Investigational	Primary satiety endpoint not met; promising microbiota engraftment; larger studies needed

## 8. Future Directions and Research Gaps

Currently, there is still no standard role of microbiome-targeted therapies in the management of *H. pylori*-associated gastritis, use in premalignant lesions, or in gastric cancer. They are under investigation as potential therapies to restore microbial balance and decrease inflammation. Despite their evolving role in the improvement of eradication rates and impacting cancer risk, together with standard treatment, especially with probiotics and FMT, there remains no recommendation to incorporate them, and there is a need for more robust RCTs [19,93,94,98,104]. Allogenic FMT also showed an improvement in the survival rates in patients with metastatic HER2-ve gastric adenocarcinoma [99]. Further larger RCTs are still needed to address that. Integrating microbiome-targeted therapies together with the development of biomarkers and providing risk stratification modules appears to be the focus and is being increasingly recognized as essential in the eradication of *H. pylori* as well as gastric cancer, as there is a complex interplay between *H. pylori*; gastric microbiota; and host demographic, clinical, and histopathological factors. As it stands, *H. pylori* remains the primary carcinogenic driver; however, gastric dysbiosis involving other bacterial taxa contributes to carcinogenesis, treatment failure, and the need for other multimodal interventional protocols [17,105].

Biomarker development is advancing, with high-throughput sequencing enabling identification of microbial signatures, such as increased Bacillus and decreased Rhizobiales, Weeksellaceae, and Veillonella. This can stratify the risk for progression from atrophic gastritis to gastric cancer. These biomarkers may allow for more precise surveillance and more targeted prevention and treatment strategies [101]. There still needs to be more consensus on the methodology for how to incorporate the use of biomarkers with the microbiome-targeted therapies for potential integration into clinical practice in the future. Clinical translation faces challenges, including but not limited to inter-individual variability in microbiome composition, lack of standardized protocols for sampling and analysis, and insufficient validation of microbial biomarkers in diverse populations. Additionally, the causal relationships between specific taxa and carcinogenesis remain incompletely defined, and regulatory, safety, and ethical considerations for interventions like FMT are unresolved.

## 9. Conclusions

Gastric cancer arises from a complex interplay between microbial, environmental, and host factors, with the gastric and intestinal microbiota emerging as critical modulators of disease risk and progression. While *Helicobacter pylori* infection remains the central carcinogenic driver, its effects are shaped by broader microbial community dynamics, including changes in bacterial diversity and shifts in specific taxa that influence inflammation, epithelial signaling, and tumorigenesis. Evidence from both human and mouse studies indicates that the presence of a conventional gut microbiota can exacerbate *H. pylori*-induced gastric pathology, highlighting the importance of microbe–microbe interactions in disease pathogenesis. Distinguishing the contributions of gastric resident versus intestinal microbiota offers clinically relevant perspectives: gastric communities may directly influence the mucosal environment, whereas intestinal microbiota may affect systemic inflammatory and metabolic pathways that indirectly modulate GC risk [106]. Clinically, these insights suggest that practitioners should consider not only eradication of *H. pylori* but also the potential consequences of microbiota disruption and the opportunities for microbiome-targeted interventions, such as probiotics, dietary modulation, or microbiota transplantation, particularly in high-risk individuals or those with pre-neoplastic lesions [107]. Future research should prioritize longitudinal cohort studies tracking microbiome dynamics before and after *H. pylori* eradication, mechanistic studies to clarify causal relationships, and trials that evaluate how microbiota-targeted therapies influence clinical outcomes. Importantly, advances in multi-omics profiling could facilitate the development of microbial signatures for risk stratification, early detection, and treatment monitoring, offering a realistic path toward integrating microbiome assessment into GC risk models. By harnessing the microbiome as both a biomarker and a therapeutic target, the field may move toward more personalized, microbiota-informed strategies that complement existing clinical practice and ultimately improve patient outcomes.

## Figures and Tables

**Figure 1 cancers-17-02795-f001:**
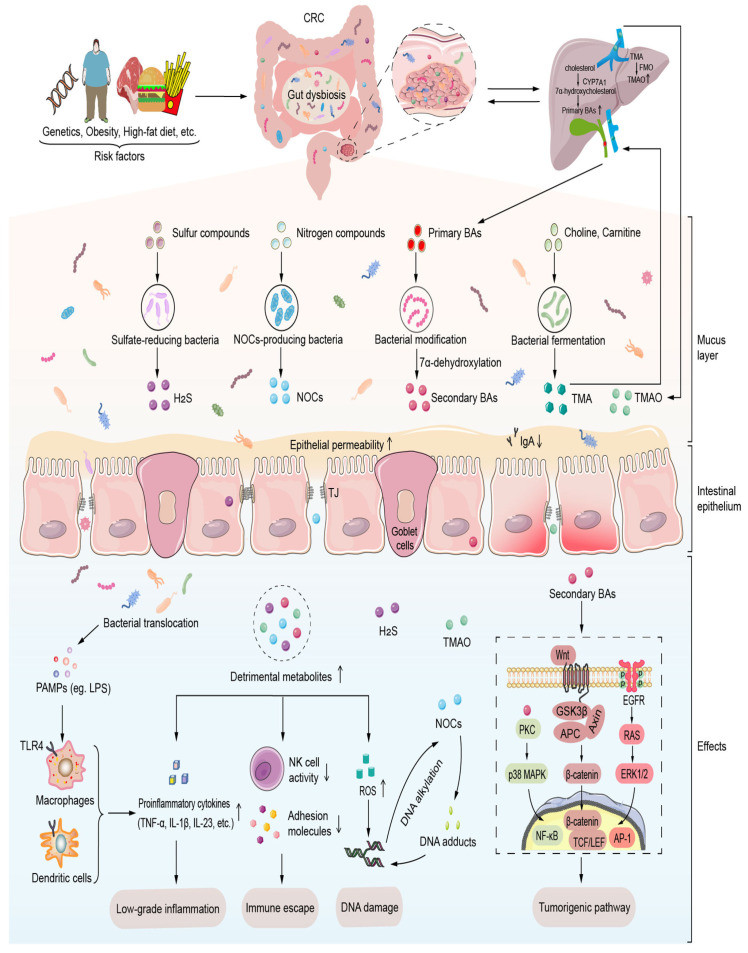
Microbial dysbiosis-driven carcinogenic mechanisms in the gut [24]. AP-1, activator protein-1; APC, adenomatous polyposis coli; BAs, bile acids; CRC, colorectal cancer; CYP7A1, cholesterol 7 α hydroxylase; EGFR, epidermal growth factor receptor; ERK1/2, extracellular signal-regulated kinase 1/2; FMO, flavin monooxygenase; GSK3β, glycogen synthase kinase 3β; H2S, hydrogen sulfide; IL-1β, interleukin-1β; IL-23, interleukin-23; LEF, lymphatic enhancement factor; LPS, lipopolysaccharide; NF-κB, factor-Kappa B; NOCs, N-nitroso compounds; p38 MAPK, p38 mitogen-activated protein kinase; PAMPs, pathogen-associated molecular patterns; PKC, protein kinase C; ROS, reactive oxygen species; TCF, T-cell factor; TJ, tight junctions; TLR, Toll-like receptor; TMA, trimethylamine; TMAO, trimethylamine-N-oxide; TNF-α, tumor necrosis factor-α.

**Table 1 cancers-17-02795-t001:** Summary of microbial genera/groups implicated in gastric cancer, their pathogenic roles, and clinical relevance.

Microbial Genus/Group	Role in Gastric Cancer Pathogenesis	Clinical Relevance
*H. pylori* [40,53,54,55,56]	Dominates gastric microbiota; triggers Correa’s cascade; induces chronic inflammation, DNA damage, and epigenetic alterations, driving tumorigenesis.	Primary carcinogenic driver; eradication reduces gastric cancer risk but may cause dysbiosis, impacting long-term outcomes.
Firmicutes (e.g., *Streptococcus*, *Lactobacillus*) [53,54,57]	Reduced abundance during *H. pylori* infection; some species are linked to precancerous lesions and sustained inflammation.	Changes in abundance may signal premalignant conditions; some strains are considered for probiotic therapy.
Bacteroidetes [53,57]	Decreased post-*H. pylori* infection and eradication; plays role in maintaining microbial balance and gut homeostasis.	Loss linked to dysbiosis and gastrointestinal symptoms after eradication therapy.
Actinobacteria (e.g., *Bifidobacterium*, *Actinomyces*) [53,54,57]	Decreased after eradication therapy; protective role in gut homeostasis; implicated in precancerous lesion development.	Protective taxa that may be targeted to restore balance post-eradication; potential probiotic candidates.
Proteobacteria (e.g., *Enterobacteriaceae*, *Escherichia-Shigella*) [54,57]	Transiently increases after eradication; associated with dysbiosis and pro-inflammatory states.	Increased abundance linked to mucosal inflammation and potential carcinogenic microenvironment.
Clostridium [50]	Produces genotoxic secondary bile acids (e.g., deoxycholic acid) that induce DNA damage and promote tumor growth.	Potential biomarker for colorectal and gastric tumorigenesis; possible target for metabolite-based therapies.
Hydrogen sulfide-producing bacteria [51]	Generate hydrogen sulfide, impairing DNA repair mechanisms and facilitating mutagenesis.	Metabolic byproducts contribute to carcinogenesis; inhibitors may reduce genotoxic effects.
Short-chain fatty acid producers (e.g., butyrate producers) [52]	Usually anti-inflammatory, but under hypoxic or oncogenic conditions, it may promote tumor cell proliferation.	Complex role; modulation may have therapeutic potential depending on tumor microenvironment context.
Streptococcus, Prevotella, Lactobacillus [54,57]	Associated with gastric mucosal changes resembling precancerous conditions post-*H. pylori* eradication.	May serve as markers for monitoring premalignant progression; some species have been explored as probiotics.
Bacillus [54]	Increased abundance correlates with progression from atrophic gastritis to gastric cancer.	Included in predictive microbial biomarker panels for neoplastic progression risk stratification.
Parvimonas, Peptostreptococcus, Rothia, Granulicatella [57]	Implicated in the development of precancerous gastric lesions.	Potential contributors to carcinogenesis; further study needed for clinical application as biomarkers or targets.

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
