# Peer review of "The Gut Microbiome’s Impact on the Pathogenesis and Treatment of Gastric Cancer—An Updated Literature Review"

_cancers, 2025, doi:10.3390/cancers17172795_

Round 1
Reviewer 1 Report
Comments and Suggestions for Authors
The paper is well organized and comprehensive, addressing the key areas such as microbial dysbiosis, immune modulation, carcinogenic metabolites, the role of H. pylori, post-eradication factors, effects of chemotherapy, and new therapies including probiotics, FMT, and personalized microbiome therapy. However i do have some concerns here:
- Some points (the role of H. pylori, inflammation-induced carcinogenesis) are duplicated within sections. Replacing duplicated content with concise ones would enhance readability.
- The majority of studies are described without criticism. Little mention is given to limitations within methodologies, data heterogeneity, or contradictory results.
- H. pylori remains pivotal to GC, but the manuscript overemphasizes it even in sections meant to emphasize non-H. pylori microbiota.
- The review can be made more readable using further figures or diagrams illustrating mechanisms, microbial profiles, or therapeutic pathways.
-No suggestion is made regarding how literature was selected (databases accessed, search terms, inclusion/exclusion criteria), lowering reproducibility and credibility as a review.
- Evidently describe how the literature was selected, including databases, keywords, and inclusion/exclusion criteria.
- Include a table that contrasts key microbial genera involved in GC, mechanisms (e.g., metabolite generated, affected pathway), and clinical relevance.
- State methodological defects or study quality cited briefly, especially for treatments like FMT and probiotics.
- Enlighten how outcomes can vary based on geographic microbiota community structure, diet, and lifestyle—especially for microbiome studies.
- Summarize recent or current cohort studies that track microbiome changes prior to and after eradication over time.
- Discuss how host genetic background may influence microbiome structure and gastric cancer susceptibility.
Author Response
Reviewer 1:
Comment 1: Some points (the role of H. pylori, inflammation-induced carcinogenesis) are duplicated within sections. Replacing duplicated content with concise ones would enhance readability.
Response 1: Thank you for the feedback. That was addressed
Comment 2: - The majority of studies are described without criticism. Little mention is given to limitations within methodologies, data heterogeneity, or contradictory results.
Response 2: Studies were described more, with criticism and limitations mentioned especially in the emerging therapeutics section.
Comment 3: - H. pylori remains pivotal to GC, but the manuscript overemphasizes it even in sections meant to emphasize non-H. pylori microbiota.
Response 3: While still retaining H pylori’s role as the main carcinogen, changes however have been made by understanding gut microbiome–gastric cancer interactions and acknowledging the H. pylori–induced changes in gastric physiology and immunity that set the stage for dysbiosis, both locally and distally in the intestinal tract. Mentions of studies that demonstrate that H. pylori infection alters not only the gastric microbiota, but also the composition and metabolic output of the intestinal microbiome, influencing systemic inflammatory and metabolic pathways relevant to cancer progression. Also a table was formulated summarizing microbial genera/groups implicated in gastric cancer, their pathogenic roles, and clinical relevance.
Comment 4: - The review can be made more readable using further figures or diagrams illustrating mechanisms, microbial profiles, or therapeutic pathways.
Response 4: 2 tables, one describing the microbiota genus and another one that goes over probiotics, prebiotics, synbiotics and FMT studies.
Comment 5: -No suggestion is made regarding how literature was selected (databases accessed, search terms, inclusion/exclusion criteria), lowering reproducibility and credibility as a review.
- Evidently describe how the literature was selected, including databases, keywords, and inclusion/exclusion criteria.
Response 5: A full paragraph about methodology after the introduction session was added.
Comment 6: - Include a table that contrasts key microbial genera involved in GC, mechanisms (e.g., metabolite generated, affected pathway), and clinical relevance.
Response 6: Table added
Comment 7: - State methodological defects or study quality cited briefly, especially for treatments like FMT and probiotics.
Response 7: Studies described more stating the types of studies and limitations in the probiotics and FMT section
Comment 8: - Enlighten how outcomes can vary based on geographic microbiota community structure, diet, and lifestyle—especially for microbiome studies.
Response: A paragraph was added mentioning that despite diet and lifestyle play role in gut dysbiosis, this still remains an underexplored area.
Comment 9: - Summarize recent or current cohort studies that track microbiome changes prior to and after eradication over time.
Response: Gastric as well as intestinal microbiome changes discussed prior to and after eradication
Comment 10: - Discuss how host genetic background may influence microbiome structure and gastric cancer susceptibility.
Response: Host genetic background paragraph added

Reviewer 2 Report
Comments and Suggestions for Authors
Dear authors,
Thank you for the opportunity to review the article "The Gut Microbiome's Impact on the Pathogenesis and Treatment of Gastric Cancer—An Updated Literature Review." This work makes a significant contribution to the systematization of knowledge in the field of gastric cancer research and presents a comprehensive, well-structured analysis of the current state of research on this topic. The review’s logical and coherent structure facilitates the reader’s understanding and enhances clarity on the issues discussed. It thoroughly covers the latest advancements and outlines prospects for future research. The presentation style is scientific yet accessible, making the article a valuable resource for a broad audience. However, I have a few questions and comments:
- Please check and correct the list of references, there are duplicate sources:
Lines 909-911 - 72. Ford AC, Yuan Y, Park JY, Forman D, Moayyedi P. Eradication therapy to prevent gastric cancer in Helicobacter pylori-positive individuals: systematic review and meta-analysis of randomized controlled trials and observational studies. Gastroenterology. 2025 Jan 15:S0016-5085(25)00041-1. doi:10.1053/j.gastro.2024.12.033. Epub ahead of print. PMID: 39824392.
Lines 945-947 - 87. Ford AC, Yuan Y, Park JY, Forman D, Moayyedi P. Eradication therapy to prevent gastric cancer in Helicobacter pylori–positive individuals: systematic review and meta-analysis of randomized controlled trials and observational studies. Gastroenterology. 2025 Jan 15. doi:10.1053/j.gastro.2024.12.033. Epub ahead of print
Lines 971-972 – 99. Wu M, Tian C, Zou Z, Jin M, Liu H. Gastrointestinal microbiota in gastric cancer: potential mechanisms and clinical applications—a literature review. Cancers (Basel). 2024;16:3547. doi:10.3390/cancers16203547
Lines 991-992 - 108. Wu M, Tian C, Zou Z, Jin M, Liu H. Gastrointestinal Microbiota in Gastric Cancer: Potential Mechanisms and Clinical Applications-A Literature Review. Cancers (Basel). 2024 Oct 21;16(20)
- The purpose of the review is missing.
- Lines 29-33 “The study specifically examined records from the previous decade, providing a detailed review of trends and patterns between 2010 and 2019, and predicts that the incidence of GC is on a decreasing trend in the majority of countries, including high-incidence countries such as Japan (ASR 36 in 2010 vs. ASR 30 in 2035) but also lowincidence countries such as Australia (ASR 5.1 in 2010 vs. ASR 4.6 in 2035).” - It is required to decipher the abbreviations ASR at the first mention in the text.
- Line 46 - UI – It is required to decipher the abbreviation at the first mention in the text.
- Line 57 – “…Helicobacter pylori infection (H. pylori).” – The full and abbreviated name of the microorganism should be written in italics. Please correct this throughout the review.
- Figure 1 – it is necessary to add a transcript of all abbreviations in the diagram in the caption to the figure.
- In your review «The Gut Microbiome’s Impact on the Pathogenesis and Treatment of Gastric Cancer—An Updated Literature Review» a whole section titled “3. Impact of Helicobacter pylori and Beyond” is devoted to the study of Helicobacter pylori. Considering that this bacterium colonizes the gastric epithelium, not the intestine, the question arises: is it justified to devote so much attention to it in a review focused on the gut microbiome? Undoubtedly, Helicobacter pylori affects the microbial balance of the entire gastrointestinal tract, but is it worth emphasizing it, given that it is not part of the gut microbiome? In my opinion, your review should pay more attention specifically to studies dedicated to the gut microbiome, comparing the microbiomes of healthy individuals and those with gastric cancer. The information on Helicobacter pylori should be rewritten with a greater focus on the interaction of this carcinogenic factor with the gut microbiome. And the information about the connection of Helicobacter pylori with other diseases (not related to gastric cancer: lines 194–210), in my opinion, is unnecessary and unrelated to the topic of the current review. In the section “3. Impact of Helicobacter pylori and Beyond” you mention the gastric microbiota (lines 212–214), however, the text contains information devoted exclusively to Helicobacter pylori and does not mention other members of the gastric microbial community.
- Lines 190-193 –“ Mouse studies provide mechanistic support for this synergy: germ-free mice colonized solely with H. pylori develop less severe gastric pathology compared to those harboring both H. pylori and a conventional microbiota . This suggests that the pathogenicity of H. pylori is amplified by microbial community dynamics [27].” – please specify which microbial community are you talking about? Do you mean the microbial communities of the stomach or intestines?
- In section “7. Emerging Therapeutic Perspectives” – In my opinion, it should be clarified that the use of probiotics and prebiotics in the fight against stomach cancer is not a completely new technology, but this area is actively developing today and is being studied as a promising element of complex treatment.
- The conclusion does not fully reflect the summary conclusions of the review on «The Gut Microbiome’s Impact on the Pathogenesis and Treatment of Gastric Cancer—An Updated Literature Review». There is also confusion with microbiomes: Helicobacter pylori is not a member of the intestinal bacterial community, and the sentence in lines 748-750 “While H. pylori remains the central microbial factor, the broader microbial ecosystem’s role is increasingly recognized in both promoting and potentially preventing gastric cancer ” - confusing, it should be reformulated.
Author Response
Reviewer 2:
Comment 1: Please check and correct the list of references, there are duplicate sources:
- Lines 909-911 - 72. Ford AC, Yuan Y, Park JY, Forman D, Moayyedi P. Eradication therapy to prevent gastric cancer in Helicobacter pylori-positive individuals: systematic review and meta-analysis of randomized controlled trials and observational studies. Gastroenterology. 2025 Jan 15:S0016-5085(25)00041-1. doi:10.1053/j.gastro.2024.12.033. Epub ahead of print. PMID: 39824392.
- Lines 945-947 - 87. Ford AC, Yuan Y, Park JY, Forman D, Moayyedi P. Eradication therapy to prevent gastric cancer in Helicobacter pylori–positive individuals: systematic review and meta-analysis of randomized controlled trials and observational studies. Gastroenterology. 2025 Jan 15. doi:10.1053/j.gastro.2024.12.033. Epub ahead of print
- Lines 971-972 – 99. Wu M, Tian C, Zou Z, Jin M, Liu H. Gastrointestinal microbiota in gastric cancer: potential mechanisms and clinical applications—a literature review. Cancers (Basel). 2024;16:3547. doi:10.3390/cancers16203547
- Lines 991-992 - 108. Wu M, Tian C, Zou Z, Jin M, Liu H. Gastrointestinal Microbiota in Gastric Cancer: Potential Mechanisms and Clinical Applications-A Literature Review. Cancers (Basel). 2024 Oct 21;16(20)
Response 1: Duplicates addressed and corrected
Comment 2: The purpose of the review is missing.
Response 2: The purpose of this review is to synthesize current evidence on the role of both gastric and intestinal microbiota in gastric cancer pathogenesis, with particular attention to the interactions between Helicobacter pylori and other microbial communities. By integrating findings from human and animal studies, we aim to clarify how microbial dysbiosis influences tumor initiation, progression, and therapeutic response. This review also evaluates emerging diagnostic and therapeutic strategies, including microbiome profiling, as potential tools for risk assessment, early detection, and personalized interventions in gastric cancer.
Comment 3: Lines 29-33 “The study specifically examined records from the previous decade, providing a detailed review of trends and patterns between 2010 and 2019, and predicts that the incidence of GC is on a decreasing trend in the majority of countries, including high-incidence countries such as Japan (ASR 36 in 2010 vs. ASR 30 in 2035) but also low incidence countries such as Australia (ASR 5.1 in 2010 vs. ASR 4.6 in 2035).” - It is required to decipher the abbreviations ASR at the first mention in the text.
Response 3: Abbreviations deciphered
Comment 4: Line 46 - UI – It is required to decipher the abbreviation at the first mention in the text.
Response 4: Abbreviation deciphered
Comment 5: Line 57 – “…Helicobacter pylori infection (H. pylori).” – The full and abbreviated name of the microorganism should be written in italics. Please correct this throughout the review.
Response 5: Feedback appreciated and addressed throughout.
Comment 6: Figure 1 – it is necessary to add a transcript of all abbreviations in the diagram in the caption to the figure.
Response 6: Feedback appreciated and addressed.
Comment 7: In your review «The Gut Microbiome’s Impact on the Pathogenesis and Treatment of Gastric Cancer—An Updated Literature Review» a whole section titled “3. Impact of Helicobacter pylori and Beyond” is devoted to the study of Helicobacter pylori. Considering that this bacterium colonizes the gastric epithelium, not the intestine, the question arises: is it justified to devote so much attention to it in a review focused on the gut microbiome? Undoubtedly, Helicobacter pylori affects the microbial balance of the entire gastrointestinal tract, but is it worth emphasizing it, given that it is not part of the gut microbiome? In my opinion, your review should pay more attention specifically to studies dedicated to the gut microbiome, comparing the microbiomes of healthy individuals and those with gastric cancer. The information on Helicobacter pylori should be rewritten with a greater focus on the interaction of this carcinogenic factor with the gut microbiome. And the information about the connection of Helicobacter pylori with other diseases (not related to gastric cancer: lines 194–210), in my opinion, is unnecessary and unrelated to the topic of the current review. In the section “3. Impact of Helicobacter pylori and Beyond” you mention the gastric microbiota (lines 212–214), however, the text contains information devoted exclusively to Helicobacter pylori and does not mention other members of the gastric microbial community.
Response 7: While still retaining H pylori’s role as the main carcinogen, changes however have been made by understanding gut microbiome–gastric cancer interactions and acknowledging the H. pylori–induced changes in gastric physiology and immunity that set the stage for dysbiosis, both locally and distally in the intestinal tract. Mentions of studies that demonstrate that H. pylori infection alters not only the gastric microbiota, but also the composition and metabolic output of the intestinal microbiome, influencing systemic inflammatory and metabolic pathways relevant to cancer progression. Also a table was formulated summarizing microbial genera/groups implicated in gastric cancer, their pathogenic roles, and clinical relevance.
Comment 8: Lines 190-193 –“ Mouse studies provide mechanistic support for this synergy: germ-free mice colonized solely with H. pylori develop less severe gastric pathology compared to those harboring both H. pylori and a conventional microbiota . This suggests that the pathogenicity of H. pylori is amplified by microbial community dynamics [27].” – please specify which microbial community are you talking about? Do you mean the microbial communities of the stomach or intestines?
Response 8: Thanks for the feedback on this. That was specified and addressed
Comment 9: In section “7. Emerging Therapeutic Perspectives” – In my opinion, it should be clarified that the use of probiotics and prebiotics in the fight against stomach cancer is not a completely new technology, but this area is actively developing today and is being studied as a promising element of complex treatment.
Response 9: Thanks for your feedback. That was addressed and stated that probiotics and prebiotics use is not new, however it still remains mostly investgational a vigorous area of research
Comment 10: The conclusion does not fully reflect the summary conclusions of the review on «The Gut Microbiome’s Impact on the Pathogenesis and Treatment of Gastric Cancer—An Updated Literature Review». There is also confusion with microbiomes: Helicobacter pylori is not a member of the intestinal bacterial community, and the sentence in lines 748-750 “While H. pylori remains the central microbial factor, the broader microbial ecosystem’s role is increasingly recognized in both promoting and potentially preventing gastric cancer ” - confusing, it should be reformulated.
Response 10: Conclusion fully reformulated, correcting the misconception that H Pylori is an intestinal bacterial community and further summarizing interactions between H pylori and different microbiomes, providing recommendations as well as takeaway points for practitioners

Reviewer 3 Report
Comments and Suggestions for Authors
The paper sounds very interesting and can be forwarded to the next step based on the editor's decision. However, the authors need to revise the paper based on the following major comments:
- The title should be more focused on the main topic of the paper.
-
- Over-Reliance on Descriptive Summarization
- Issue: The manuscript often summarizes studies without critical appraisal or synthesis.
- Recommendation: The authors should compare findings across studies, highlight controversies or limitations (e.g., regional variation, confounders in observational studies), and point out where evidence is inconclusive.
- Lack of Visual Summaries
- Issue: Mechanisms of microbiota-induced carcinogenesis are described in detail but would benefit from additional figures or summary diagrams.
- Recommendation: Add a visual summary (e.g., flowchart or infographic) to depict:
- Dysbiosis → inflammation → signaling → carcinogenesis
- Post-H. pylori eradication changes
- Summary of current therapeutic approaches
- Therapeutic Section Needs Refinement
- Issue: While probiotics, prebiotics, synbiotics, and FMT are well covered, the section lacks differentiation between clinically proven and hypothesis-generating approaches.
- Recommendation: Include a table summarizing clinical trial findings, distinguishing between preclinical, Phase I/II trials, and meta-analyses. Clearly label what is evidence-based vs. investigational.
- Writing Style and Clarity
- Issue: Some sections are verbose and would benefit from tighter editing.
- Examples:
- Repetition of points on residual risk post-H. pylori eradication
- Redundant citations in consecutive sentences
- Recommendation: Shorten and consolidate overlapping paragraphs, especially in Sections 5 and 9.
- Conclusion Could Be More Impactful
- Issue: The conclusion restates rather than synthesizes findings.
- Recommendation: Use this section to provide clear takeaways:
- What clinical changes should current practitioners consider?
- Where is further research most urgently needed?
- The following papers can be added to the current article review:
- 1: 10.1186/s13099-025-00729-w
- 2: https://doi.org/10.3390/app122312474
- 3: 10.2147/CEG.S243337
Send the paper to a professional English editor.
Author Response
Reviewer 3: The manuscript often summarizes studies without critical appraisal or synthesis.
Response 1: Studies were described more with mentions of methodologies, criticism and limitations mentioned especially in the emerging therapeutics section.
Comment 2: Lack of Visual Summaries
Issue: Mechanisms of microbiota-induced carcinogenesis are described in detail but would benefit from additional figures or summary diagrams.
Recommendation: Add a visual summary (e.g., flowchart or infographic) to depict:
Dysbiosis → inflammation → signaling → carcinogenesis
Response 2: A diagram depicting dysbiosis resulting in inflammation, signaling and carcinogenesis is in the manuscript. Two tables were added addressing the microbiome, role in gastric carcinogenesis and clinical relevance if any.
Comment 3:
Post-H. pylori eradication changes
Summary of current therapeutic approaches
Therapeutic Section Needs Refinement
Issue: While probiotics, prebiotics, synbiotics, and FMT are well covered, the section lacks differentiation between clinically proven and hypothesis-generating approaches.
Recommendation: Include a table summarizing clinical trial findings, distinguishing between preclinical, Phase I/II trials, and meta-analyses. Clearly label what is evidence-based vs. Investigational.
Response 3: Clinical trial findings and evidence level summarized in a table with added text providing criticism to studies especially in probiotics and FMT in terms of limitations and small population size
Comment 4: Writing Style and Clarity
Issue: Some sections are verbose and would benefit from tighter editing.
Examples:
Repetition of points on residual risk post-H. pylori eradication
Redundant citations in consecutive sentences
Recommendation: Shorten and consolidate overlapping paragraphs, especially in Sections 5 and 9.
Response 4: Adjustments are done to reduce redundance in text.
Comment 5:
Conclusion Could Be More Impactful
Issue: The conclusion restates rather than synthesizes findings.
Recommendation: Use this section to provide clear takeaways:
What clinical changes should current practitioners consider?
Where is further research most urgently needed?
How might microbiome profiling realistically be integrated into GC risk assessment?
The following papers can be added to the current article review:
1: 10.1186/s13099-025-00729-w
2: https://doi.org/10.3390/app122312474
3: 10.2147/CEG.S243337
Response 5:
Conclusion was edited to provide clear takeaway points with including the trials mentioned in the conclusion.

Round 2
Reviewer 1 Report
Comments and Suggestions for Authors
Thank you for answering my questions. good luck
Reviewer 3 Report
Comments and Suggestions for Authors
The authors revised the paper based on my major comments.
One last issue is the English editing. Please send the paper to a professional English editor.
Comments on the Quality of English LanguageThe authors revised the paper based on my major comments.
One last issue is the English editing. Please send the paper to a professional English editor.